# Chemical Components of *Oxytropis pseudoglandulosa* Induce Apoptotic-Type Cell Death of Caco-2 Cells

**DOI:** 10.3390/molecules27144609

**Published:** 2022-07-19

**Authors:** Tuya Narangerel, Małgorzata Zakłos-Szyda, Michał Sójka, Iwona Majak, Maria Koziołkiewicz, Joanna Leszczyńska

**Affiliations:** 1Institute of Natural Products and Cosmetics, Faculty of Biotechnology and Food Sciences, Lodz University of Technology, Stefanowskiego 2/22, 90-537 Lodz, Poland; nartuya11@gmail.com; 2Institute of Molecular and Industrial Biotechnology, Faculty of Biotechnology and Food Sciences, Lodz University of Technology, Stefanowskiego 2/22, 90-537 Lodz, Poland; malgorzata.zaklos-szyda@p.lodz.pl (M.Z.-S.); maria.koziolkiewicz@p.lodz.pl (M.K.); 3Institute of Food Technology and Analysis, Faculty of Biotechnology and Food Sciences, Lodz University of Technology, Stefanowskiego 2/22, 90-537 Lodz, Poland; michal.sojka@p.lodz.pl (M.S.); iwona.majak@p.lodz.pl (I.M.)

**Keywords:** *Oxytropis*, kaempferol glycosides, herbs

## Abstract

*Oxytropis pseudoglandulosa* plant is used in traditional Mongolian medicine. However, its chemical composition and biological properties are poorly explored. In this study, the total content of polyphenols and flavonoids as well as antioxidant activity were verified in plant extract. The total phenolic and flavonoid contents were determined by spectrometric (6.62 mg GAE/g and 10.32 mg QE/g) and chromatographic (17,598 mg/kg and 17,467 mg/kg) assays. The antioxidant potential was investigated by DPPH assay and yielded IC_50_ at 18.76 µg/mL. Twelve phenolic compounds were identified as components of *O. pseudoglandulosa* extract. Kaempferol-3-*O*-robinosyl-7-*O*-rhamnoside and kaempferol-3-(*p*-coumaroyl)-rutinosyl-7-rhamnoside made up 80% of determined components and were found to be the major polyphenolic compounds. The biological properties of *O. pseudoglandulosa* extracts were determined in vitro using human epithelial adenocarcinoma Caco-2 cell line. Low concentrations of extract (0–30 µg/mL) exhibited protective effects against cell damage caused by chemically induced oxidative stress. Elevated concentrations, on the other hand, resulted in apoptotic-type cell death induction. Metabolic failure, ROS elevation and membrane permeabilization observed in cells upon incubation with extract dosages above 50 µg/mL allowed us to conclude on *O. pseudoglandulosa* being predominantly a necrosis inducer.

## 1. Introduction

Civilizational diseases, especially diet-related ones, pose a major medical and social challenge in developed countries. This includes atherosclerosis and cardiovascular diseases, neoplastic diseases, neurodegenerative diseases and food allergies. Development of these conditions is likely a consequence of inflammatory processes related to the excessive production of free radicals in the human body. Imbalanced nutrition, mostly associated with the consumption of highly processed foods is at the root of this phenomenon [1]. Researchers aim to address this issue with novel food supplements devised to not only act as preservatives, but at the same time exert health-promoting effects. Herbal components, many of which are still to be identified, are suitable candidates for this role [2,3,4]. The use of herbs as food additives dates back to antiquity. Apart from their taste improving and shelf-life extending properties, they can also work as natural remedies by relieving certain symptoms through their anti-inflammatory, antiseptic, immunosuppressive effects, etc. [5]. This is especially true for plants containing numerous bioactive compounds as their inherent chemical components. Additionally, there is growing evidence that increased intake of fruits, vegetables or herbs reduces the risk of human cancer development [6]. The observed relationship is very often associated with chemoprotective effect of phytocompounds, capable of counteracting DNA and protein damage, as well as with an induction of apoptosis. Given how many *Oxytropis* plants are listed as forms of treatment to various conditions, including inflammation, cold, swelling and bleeding, the understudied *Oxytropis pseudoglandulosa*, widely used in traditional Mongolian medicine, was recognized as a promising candidate for detailed research.

In a previous work [7] we presented the chemical composition of the essential oil and lipid extracts of *O. pseudoglandulosa* plant species, as well as the total phenolics and flavonoids content, antimicrobial properties and allergenicity. This work focuses on the biological properties of *O. pseudoglandulosa*. The plant was hence subjected to spectrophotometric assay and LC-MS chromatography to point out chemical components potentially responsible for its reported medicinal effects. The biological properties of plant extract obtained from *O. pseudoglandulosa* were determined through in vitro studies performed with animal cell cultures. As intestinal cells are typically exposed to large quantities of dietary compounds, human epithelial adenocarcinoma Caco-2 cell line was chosen as a model cellular target. These cells are commonly used for in vitro studies on the intestinal absorption mechanisms, as well as other biological processes, including cytoprotection or cellular death induction [8]. Cytotoxicity of the extract was determined with PrestoBlue assay, whereas its cytoprotective activity was studied in oxidative stress conditions induced by *tert*-butylhydroperoxide (*t*-BOOH). To investigate the potential of *O. pseudoglandulosa* as an inducer of apoptosis/necrosis, the intracellular ATP level, mitochondrial membrane potential and activation of caspases 3/7 were tested. To the best of our knowledge, this is the first study demonstrating the cell-based in vitro activity of *O. pseudoglandulosa* extract.

## 2. Results and Discussion

### 2.1. Chemical Characterization of O. pseudoglandulosa

#### 2.1.1. Reducing Sugars, Total Phenolic Content, Total Flavonoid Content

The content of reducing sugars, expressed as glucose, in the studied plant was determined at 17.32 mg per gram of dry plant mass. The total phenolic content (TPC) was 6.62 mg of gallic acid equivalents (GAE) per gram of dry plant mass, while total flavonoid content (TFC) was 10.32 mg of quercetin equivalents (QE) per gram of dry plant mass. To put these values into perspective, according to Berber et al., TPC of *O. halleri* was 78.84 mg GAE/g [9]. In a related study, Miladinović et al. reported TFC of approximately 0.1 mg/g as determined in the leaves of *Oxytropis pilosa* L. [10]. Higher levels of phenolic and non-phenolic flavonoid compounds in the chemical composition of a medicinal plant are recognized to be indicative of more pronounced bioactive properties and antioxidant effects [11]. These were thus elucidated in the following part of the work.

#### 2.1.2. Polyphenolic Composition

The identification of polyphenolic compounds was based on comparing their retention times, UV spectra and MS spectra with those of known standard compounds. In the case of standards being unavailable, the experimental spectral data was compared with data acquired from the literature. Twelve phenolic compounds (three of which were phenolic acids and nine were flavonols) were identified in *Oxytropis pseudoglandulosa* and shown in Table 1.

The first three compounds were identified as: protocatechuic acid hexoside, 5-caffeoylquinic acid (chlorogenic acid) and 4-caffeoylquinic acid (cryptochlorogenic acid). For the first compound, the loss of a hexose moiety (−162.05 amu) was reported in fragmentation spectrum ion at *m*/*z* 153.01, which corresponded to hexose-free protocatechuic acid molecule. Compounds no. 2 and 3 were isomers, and their ion masses in the negative ionization mode were the same. The identity of compound no. 2 (5-caffeoylquinic acid) was confirmed based on the commercial standard. The fragment ions of the compound no. 3 were matched with the 4-caffeoylquinic acid (cryptochlorogenic acid). In both compounds no. 2 and 3, the main fragment ion *m*/*z* 191 was associated with the occurrence of quinic acid moiety. The major compounds reported in *O. pseudoglandulosa* were identified as flavonol glycosides of kaempferol and quercetin. Identification of most flavonols was based on their UV spectra, MS and MS/MS spectra, i.e., accurate mass measurements of the pseudo-molecular [M − H] and their post-fragmentation pseudo-ions, as well as literature data. In the case of kaempferol-3-*O*-robinosyl-7-*O*-rhamnoside (compound no. 7) and kaempherol-3-*O*-rutinoside (compound no. 10), the identification was performed based on commercial standards. Based on scientific literature and acquired MS data, compounds no. 4, 5, 6, 8, 9 and 12 were identified as: kaempferol-3-(rhamnosyl)rutinosyl-7-rhamnoside (4), quercetin-3-*O*-rutinosyl-7-*O*-rhamnoside (5), kaempferol-3-*O*-rutinosyl-7-*O*-rutinoside (6), kaempferol-3-*O*-rutinosyl-7-*O*-rhamnoside (8), kaempferol-3-(*p*-coumaroyl,rhamnosyl)-rutinosyl-rhamnoside (9) and kaempferol-3-*O-*rhamnoside (12), respectively. In the MS^2^ fragmentation spectra of these tested compounds, a weight loss of −146.05 amu was observed in each case, except compound 6, which was related to the detachment of the rhamnosyl ion. This type of fragmentation is characteristic for three or four-glycoside flavonols, where during MS^2^ fragmentation, the cleavage of the glycosidic residue most often occurs at the 7-position [12,13]. In the compound no. 4, MS^3^ fragmentation of the pseudo-ion [M – H − 162] 739.21 indicated the presence of pseudo-ions *m*/*z* 593.16 and 285.04, which was related to the detachment of rhamnose ion (−146 amu) and rutinose (−308 amu), pointing towards its link to a non-phenolic hydroxyl. Hence, this compound was identified as kaempferol-3-(rhamnosyl)rutinosyl-7-rhamnoside. MS^2^ fragmentation of the compounds no. 5 and 8 indicated the presence of stable pseudo-ions *m*/*z* 609.15 and 593.15, respectively, which was related to the presence of rutinosyl residues linked to the quercetin and kaempferol aglycone, respectively. The compound no. 6 was characterized by atypical MS^2^ fragmentation: for this compound the loss of the *m*/*z* 308 ion (MS^2^ [M − H] → 593) was observed, which was thought to be related to the loss of the rutinosyl residue in position 7. Hence, this compound was identified as kaempferol-3-rutinosyl-7-rutinoside. The compound no. 7 was identified through comparison with the standard, kaempferol-3-*O*-robinosyl-7-*O*-rhamnoside. Although it showed similarity to the compound no. 8, the latter was identified as kaempferol-3-*O*-rutinosyl-7-*O*-rhmnoside. The compound no. 9 with the largest molecular mass was identified as kaempferol-3-(*p*-coumaroyl,rhamnosyl)-rutinosyl-rhamnoside. The use of the Q Exactive Orbitrap detector made it possible to determine the exact mass (−146.0378 amu) of the pseudo-ion detached in the MS^2^ fragmentation, which indicated the presence of a *p*-coumaric acid residue in the structure of this compound. Moreover, the UV-Vis spectrum of this compound was characterized by a shift of the absorbance maximum to the value of 317 nm, which is characteristic of acylated flavonols [12]. Compounds no. 10 and 11 had the same fragment ions (derived from rutinosides), but their aglycones were identified as kaempferol and isorhamnetin, respectively. Based on the report by Jang et al., compound 12 was identified as kaempferol-3-*O*-rhamnoside [14].

The content of polyphenolic compounds in *O. pseudoglandulosa* was determined using standard curves plotted for chlorogenic acid, kaemferol-3-rutinoside, and kaempferol-3-*O*-robinosyl-7-*O*-rhamnoside, as shown in Table 2.

Chromatographic quantitative analysis showed that total content of polyphenolic compounds in dry mass of *O. pseudoglandulosa* is 17,598 mg/kg. The analysis on the individual compounds indicated that the main components in this plant were kaempferol-3-*O*-robinosyl-7-*O*-rhamnoside at around 12,355 mg/kg and kaempferol-3-(rhamnosyl)-rutinosyl-7-rhamnoside at 1593 mg/kg. The content of the remaining flavonols was at a much lower level. Even though the kaempferol-3-*O*-robinosyl-7-*O*-rhamnoside was previously isolated from *O. varlacovii Serg* and *O. falcate* [15], their quantitative analysis has not appeared in scientific data.

#### 2.1.3. Antioxidant Properties

The antioxidant effect exerted by *O. pseudoglandulosa* was tested and given as IC_50_ value expressed in micrograms of Trolox equivalents per mL of methanol extract. The IC_50_ value of the plant extract sample was determined with DPPH method and equaled 18.76 µg/mL. The result was comparable to the one determined in parallel using Trolox (19.37 µg/mL) as a standard antioxidant. Based on this finding, *O. pseudoglandulosa* is thought to have a high antioxidant effect. No comparative reports can be referenced since the antioxidant properties and chemical composition of *O. pseudoglandulosa* have not yet been described. Only partial characteristics of the *Oxitropus falcate* Bunge plant material was carried out by Jiang et al. (2008), who determined the IC_50_ of 2.05 µg/mL in ethyl acetate plant extracts using DPPH method [16].

The radical scavenging activities of five individual compounds isolated from the researched plant were also measured. The results showed that kaempferol (IC_50_ = 110 µg/mL), rhamnetine (IC_50_ = 140 µg/mL) and rhamnocitrin (IC_50_ = 150 µg/mL) had significant antioxidant activities. On the other hand, the antioxidant activities of the two dihydrochalcones (2′, 4′-dihydroxychalcone and 2′, 4′, β-trihydroxy-dihydrochalcone) were very weak (IC_50_ = 2050 µg/mL). Although the properties of the chemical compounds have been described before, the antioxidant effect of an *Oxitropis* as a plant material has not yet been well-documented.

### 2.2. Biological Characterization of O. pseudoglandulosa

#### 2.2.1. Cytotoxic Properties of *O. pseudoglandulosa*

The effect of plant extract at the concentrations 10 to 100 µg of freeze-dried extract per mL on Caco-2 cells viability was studied upon 24 h incubation. The approach involved the use of Presto Blue^®^ reagent, which is converted by mitochondrial enzymes to fluorescent product in proportion to the number of metabolically active cells [17]. The cytotoxic effect of *O. pseudoglandulosa* extracts against Caco-2 cells increased in line with the rising extract concentration, as shown in Figure 1. It could be observed that the concentrations of the tested extracts higher than 60 µg/mL exerted the strongest metabolic inhibitory effect, and the IC_50_ concentration was equal to 65 µg/mL. Concentrations below 20 µg/mL had no statistically significant influence on metabolic activity.

#### 2.2.2. The Effect of *O. pseudoglandulosa* Extract on Intracellular Oxidative Stress

Intracellular reactive oxygen species are involved in cellular signaling, but their excessive accumulation leads to oxidative stress and results in damage to proteins and lipids. Therefore, in the next step, the influence of plant extracts on intracellular level of ROS in Caco-2 cells was determined with fluorogenic DCFH-DA probe. As shown in Figure 2, pre-incubation of cells with extracts at 0–40 µg/mL decreased intracellular ROS level by 5–25%. It is known that this type of biological activity plays a crucial role in protection against oxidation of cellular lipids, proteins and DNA. Since epithelial cells are constantly exposed to luminal oxidants from ingested foods, the observed reduction of oxidative stress indicates cytoprotective effect of *O. pseudoglandulosa*. Consequently, as a next step of research, the chemopreventive potential of the medicinal plant extract was tested against chronic oxidative stress induced by *t*-BOOH.

The reagent *t*-BOOH generates oxidative stress under in vitro conditions through formation of tert-butoxyl, peroxyl, alkoxyl and methyl radicals that catalyze lipid peroxidation, DNA strand breakage and disturbance in intracellular calcium homeostasis. As compared to the control cell culture (0% plant extract sample), *t*-BOOH caused over a 2-fold elevated level of intracellular oxidative stress. However, when cells were pre-incubated with the extract concentrations 10–40 µg/mL prior to *t*-BOOH treatment, the adverse level of oxidative stress was diminished by nearly 30%. The most pronounced protective effect against oxidative damage induced by *t*-BOOH was observed for 10 µg/mL concentration (Figure 3a). The exposure to *t*-BOOH also influenced the metabolic activity of Caco-2 cells and lowered it to almost 75%. When cells were pre-incubated with the plant extract at concentrations 10–30 µg/mL, the cytotoxic effect of *t*-BOOH on metabolic activity was attenuated by nearly 10% (Figure 3b).

Overall comparison of the results suggests that *O. pseudoglandulosa* extract used at concentrations between 10 and 30 µg/mL can be used as a cytoprotective phytocomponent against oxidative stress generated by ingested foods or intestinal microbiota. This applicability can be justified by the radical-scavenging activity of chemical components in *O. pseudoglandulosa*, most of which were phenolic compounds. The observed cytoprotective mechanism may also be associated with activation of certain intracellular antioxidant enzymes, such as glutathione peroxidase, superoxide dismutase or catalase, all of which are the first line of defense against ROS in the human body [18]. Many of the phenolic constituents identified in *O. pseudoglandulosa* extract were derivatives of kaempferol or isorhamnetin marked by different glycosylation patterns. Jung et al. demonstrated that kaempferol glycosides isolated from *Brassica juncea* were identified as DPPH scavengers [19]. A recent study also demonstrated that kaempferol exhibited stronger antioxidant activity than its respective glycosides-kaempferol-7-*O*-glucoside, kaempferol-3-*O*-rhamnoside and kaempferol-3-*O*-rutinoside [20]. A study performed on diabetic rats proved that kaempferol contributed to the antioxidant effect, as well as decreased the level of lipid peroxidation markers [21]. In a related study, isorhamnetin glycosides identified in *Opuntia ficus-indica* were shown to decrease nitric oxide production in RAW 264.7 cells, as well as COX-2, TNF-α, and IL-6 levels in a rat with ear edema inflammation [22]. Despite the fact, that protocatechuic acid concentration was determined as the lowest among identified chemicals, this compound was demonstrated as being able to attenuate oxidative damage in human umbilical vein endothelial cells (HUVECs) induced by palmitic acid [23]. Given the role ROS play in carcinogenesis, the reduction of ROS by *O. pseudoglandulosa* can potentially be a valuable tool towards anti-cancer treatment.

#### 2.2.3. The Effect of *O. pseudoglandulosa* Extract on Cell Death Induction

Comparison of data presented in Figure 1 and Figure 2 implies that *O. pseudoglandulosa* extract concentrations above 30 µg/mL can increase the ROS formation, which was accompanied by reduction in metabolic activity. In the presence of the highest extract concentrations (90–100 µg/mL) with the most effective reduction of metabolic activity, the level of generated ROS dropped down dramatically. It can be suspected that elevated extract concentrations triggered cell death. To identify the mechanism of *O. pseudoglandulosa* extract cytotoxicity, ATP level was measured in cells cultured with extracts at concentrations below IC_50_ values. Caco-2 cells treatment with the extract at 30 µg/mL resulted in ATP level decrease by 10%, and the effect grew to 50% when cells were incubated with extract at 50 µg/mL concentration (Figure 4a).

Given the fact that mitochondria are a driving force behind cellular ATP synthesis, the influence of extract on mitochondrial membrane potential (MMP) was tested with JC-1 probe, a cationic carbocyanine dye that accumulates in mitochondria. As shown in Figure 4b, *O. pseudoglandulosa* extract reduced mitochondrial membrane potential in a concentration-dependent manner. The highest efficiency was revealed at 50 µg/mL concentration with the MMP value being declined to almost 60%.

Metabolic activity is closely related to mitochondria activity and ATP level. Therefore, due to the reported depletion of ATP, decrease of MMP and increase of ROS, we aimed to examine the type of cellular death induced by elevated concentrations of the extract. To this end, the fluorescent conjugates of Annexin V were applied, which are able to bind to the externalized phosphatidylserine (PS) in the cell membrane. Translocation of PS to outer leaflet of the cell membrane is characteristic for apoptotic cells. Simultaneously, propidium iodide (PI) was used, which is able to stain nuclei of necrotic cells with permeabilized membranes. As shown in Figure 5a, the highest number of apoptotic cells positive for Annexin V staining was observed after the cells were treated with 30–40 µg/mL of *O. pseudoglandulosa* extract (about 15–25% increase). After incubation with 50 µg/mL of the extract, a prominent level of cells positive for propidium iodide staining was detected—in this case, nearly 25% more cells exhibited high extent of red fluorescence specific for necrotic nuclei.

Apoptotic programmed cell death induction is used in cancer prevention. This type of cell death is naturally implicated in the removal of defective or unwanted cells and it occurs without leakage of intracellular components nor with induction of inflammation [24]. Caspases are critical enzymes involved in apoptosis. Among these enzymes, caspase-3, -6 and -7 participate in the execution of apoptosis. Their activity is one of the hallmarks of apoptosis process, along with other phenomena, like phosphatidylserine exposure in outer cellular membrane, cleaved poly (ADP-ribose) polymerase (PARP), chromatin condensation or genomic DNA fragmentation and cell blebbing [24]. As a next step of our work, acute influence of plant extract on the activation of apoptosis-related proteinases was tested. As shown in Figure 5b, a distinct increase in caspases-3/7 activation was observed after the cells were treated with 30–40 µg/mL of the extract. Therefore, it can be assumed that concentrations between the highest non-cytotoxic (IC_0_) and IC_50_ (65 µg/mL) exhibit bioactive potential, which entails induction of apoptotic-type cell death in Caco-2 cells. Metabolic failure, ROS elevation and membrane permeabilization observed in cells incubated with the extract dosages higher than 50 µg/mL allowed to suppose that *O. pseudoglandulosa* induced predominantly necrotic-like features, which was manifested by enhanced fluorescence of PI-stained nuclei. These results may point to necrosis or regulated necroptosis mechanism of cell death when lower or moderate concentrations of plant extracts are applied [25]. Higher concentrations are thought to be cytotoxic due to excessive oxidative potential, which can impede the maintenance of plasma membrane integrity. However, this idea needs more detailed investigation.

The main components identified in *O. pseudoglandulosa* extract were phenolic compounds, among which kaempferol glycosides were shown to occur in the largest quantities. Whereas the biological significance of glycosides is not very clear, there are many studies that confirm anti-cancer potential of kaempferol in different cancerous cell lines, i.e., mouse colon cancer CT26 cell line, mouse melanoma B16F1 cell line, human hepatoma cell line HepG2 [20,26]. In the former case, the HepG2 cells exposed to kaempferol were subject to apoptosis marked by inhibition of AKT phosphorylation, cleavage of caspase-9, caspase-7, caspase-3 and PARP. What is more, comparison of the potential of glycosides and their aglycons demonstrated that, among different glycosides (kaempferol-7-*O*-glucoside, kaempferol-3-*O*-rhamnoside and kaempferol-3-*O*-rutinoside), kaempferol as the aglycone was the most potent inducer of the apoptotic type of cellular death, whereas its glycosides showed poor activity [20]. Comparable pattern of activity was observed with regard to free radical scavenging activity with the most active aglycone, which was followed by less active kaempferol-7-*O*-glucoside, kaempferol-3-*O*-rhamnoside and kaempferol-3-*O*-rutinoside with no significant activity. Recent studies demonstrated that kaempferol glycoside derivatives isolated from the aerial parts of *Lens culinaris* Medik reduced DNA damage caused by etoposide in human peripheral blood mononuclear cells (PBMCs), but had no impact on DNA damage in human leukemia HL-60 cells [27]. Additionally, kaempferol induced DNA damage in HL-60 cells, which suggests that kaempferol derivatives can be further explored as potential agents protecting normal cells against DNA damage induced by etoposide. Whereas potent biological activity of kaempferol is known, there is little knowledge about its glycosides.

There are some plants of the genus *Oxytropis* with generally demonstrated antioxidant, anti-inflammatory or anticancer activities [28], nevertheless little is known about the molecular explanations of these effects. Thus far, there has been one identified study demonstrating that extract from *O. pseudoglandulosa* was able to decrease the proliferation and migration of vascular smooth muscle cells (VMSC) via suppression of ERK1/2 and Akt signaling pathways, but without cytotoxic effect [29]. Our study presented that *O. pseudoglandulosa* extract, rich in kaempferol glycosides, revealed strong biological activity against Caco-2 cells. To the best of our knowledge, this is the first communication matching results from in vitro cell-based studies with the antioxidant properties of *O. pseudoglandulosa* extract constituents and no comparative reports can be referenced in this regard. It should be noted, though, that the reported activity was determined using the mixture of multiple kaempferol derivatives and other phenolics, which can interact between themselves and produce a distinct combination output. The presented results, however, allow us to identify *O. pseudoglandulosa* extract as a source of phytocompounds with a potential to modulate cellular signal transduction pathways. The influence of the extract used as a cytoprotectant and a regulator of cellular signaling at non-cytotoxic concentrations will be examined in more detail in future studies.

## 3. Materials and Methods

### 3.1. Standards and Reagents

DPPH, Folin–Ciocalteu reagent, gallic acid, quercetin, chlorogenic acid, 5% carbon dioxide, DMSO and Trolox (Sigma-Aldrich, Poznan, Poland); kaemferol-3-rutinoside and kaempferol-3-*O*-robinoside-7-*O*-rhamnoside (Sigma-Aldrich, Schnelldorf, Germany); gallic acid, quercetin, Trolox, methyl nonadecanoate, MTBE, TMSH, chlorogenic acid, rosmarinic acid, *t*-BOOH, DCFH-DA, JC-1 dye, CCCP (Sigma-Aldrich, Poznan, Poland); methanol, ethanol, isooctane, sodium carbonate, sodium hydroxide, aluminium chloride, sodium acetate, sulfuric acid, formic acid, acetonitrile, and resazurin (POCh Gliwice, Poland).

### 3.2. Plant Material

Aerial part of the medicinal plant *O. pseudoglandulosa* was collected in its entirety from the local market in Ulaan-Uul sum, Khuvsgul province, Ulaanbaatar, Mongolia. The material was subjected to drying in the shade at 15–17 °C for three weeks. Dr Urgamal Magsar from the Institute of General and Experimental Biology (Mongolian Academy of Sciences, Mongolia) confirmed the botanical identification of the plants. The aerial part of the medicinal plant was gently ground using a pestle and mortar and packed in PE plastic ziplock bags. Plant was always stored in a dark place at room temperature and was subjected to extraction.

### 3.3. Extraction Procedure for Chemical Analyses

The extraction of phenolic compounds was performed prior to the determination of total phenolics content, total flavonoid content, antioxidant activity and LC-MS analysis. Weighted portions of 0.50 g of finely ground sample were extracted with 12 mL of 50% methanol (*v*/*v*) acidified with 1% of formic acid (*v*/*v*) and mixed using a shaker (TTS 2, Yellow Line, IKA-Werke, Staufen, Germany) at room temperature for one hour. Methanol was acidified to avoid oxidation of phenolics compounds. The supernatant was then separated via centrifugation at 10,000 rpm, for 5 min (MPW-251, Med instrument, Warsaw, Poland) and passed through paper filters (BOECO, Hamburg, Germany, Grade 3 hw). The sediment was re-extracted with 10 mL of the same extractant and the separation process was repeated. Finally, collected supernatants were combined and filled up to 25 mL with 50% methanol (*v*/*v*) and kept in the freezer (−20 °C) until tested.

### 3.4. Total Phenolic Content

The total phenolic content was determined using the Folin–Ciocalteu assay. Briefly, the volume of 50 µL of 50% methanol extract was mixed with 250 µL of Foline–Ciocalteu’s reagent, allowed to react for 5 min and next 2.5 mL of 20% sodium carbonate solution was added and filled up with deionized water up to 25 mL. The mixture was incubated for one hour in a dark place. The absorbance of the mixture was measured at 720 nm using a spectrophotometer HP8453 (Hewlett Packard, Palo Alto, CA, USA/Agilent, Santa Clara, CA, USA). All samples were tested in triplicates. A standard curve plotted for gallic acid was used for the calculations. The results were expressed in mg of gallic acid equivalents (GAE) per g of dry plant mass.

### 3.5. Total Flavonoid Content

The aluminum chloride colorimetric assay was adapted for the determination of total flavonoid content. Namely, 500 µL of sample extract was combined with 1.5 mL of 80% methanol (*v*/*v*), 100 µL of 10% AlCl_3_, 100 µL of CH_3_COONa (1 M) and 3.0 mL deionized water. The incubation was performed in a dark place for 30 min. Absorbance was measured at 415 nm using a spectrophotometer HP8453 (Hewlett Packard/Agilent, USA). All samples were tested in triplicates. A standard curve plotted for quercetin was used for the calculations. Results were expressed in mg of quercetin equivalents (QE) per g of dry plant mass.

### 3.6. Free Radical Scavenging Activity

First, 3 mL of 0.1 mM of DPPH solution diluted with 80% methanol (*v*/*v*) was combined with 1.0 mL of the extract. The mixture was vortexed and incubated in the dark for 30 min. Absorbance was measured at 517 nm using a UV–VIS spectrophotometer HP8453 (Hewlett Packard/Agilent, USA). As a negative control, the sample was substituted with the solution previously used for extraction. The same procedure was applied for Trolox solutions in range 5–50 µg/mL used to plot a standard curve. All analyses were performed in triplicates. The DPPH radical scavenging capacity (RSC) was calculated according to the following equation:RSC,% = 100 (A0 − A)/A0
where: A—average absorbance of the sample, A0—average absorbance of control (DPPH). In order to determine the IC_50_, the 50% inhibitory concentration was plotted against the percent of DPPH activity. The IC_50_ value was expressed in μg of Trolox equivalents (TE) per mL of methanolic extract.

### 3.7. LC-MS Analysis of Phenolic Compounds

The detailed analysis of polyphenols was carried out using a Dionex Ultimate 3000 HPLC chromatograph coupled with DAD and Q Exactive Orbitrap mass spectrometer (Thermo Fisher Scientific, Waltham, MA, USA). The operating parameters of the LC-MS system were set up as follows: column: 250 mm × 4.6 mm, 5 µm, Luna C18(2) 100 Å, column temperature: 35 °C, mobile phase flow rate: 1 mL min^−1^ and injection volume 20 µL. The mobile phases consisted of 1% formic acid (*v*/*v*) in water (solvent A) and 80:20 (*v*/*v*) acetonitrile-water (solvent B). The following gradient was used: 0–6.5 min, 5% (*v*/*v*) B; 6.5–12.5 min, 5–15% (*v*/*v*) B, 12.5–44 min, 15–45% (*v*/*v*) B, 44–45 min, 45–75% (*v*/*v*) B, 45–50 min, 75% (*v*/*v*) B, 50–52 min, 75–5% (*v*/*v*) B, and 52–65 min, 5% (*v*/*v*) B. The MS system coupled to HPLC was an Orbitrap mass spectrometer equipped with an H-ESI probe used in the negative mode. The mass detector parameters were set up as follows: vaporizer temperature: 500 °C, ion spray voltage: 4 kV, capillary temperature: 400 °C, sheath gas: 75 units, auxiliary gas: 20 units and scan range from 200 to 2000 *m*/*z*. The detector was operating in either full MS or full MS/dd-MS2 scan modes. To generate MS2 data, the full MS/dd-MS2 scan mode was applied. The collision energy used to generate MS2 spectra was set to 20. Tuning and optimization were performed using direct injection of extracts diluted in an 80:20 (*v*/*v*) mixture of mobile phases A and B at a flow rate of 0.25 mL/min. Chromatographic data of the *O. pseudoglandulosa* extract was collected using Xcalibur software v3.1 (Thermo Fisher Scientific, Waltham, MA, USA).

The standard solutions of chlorogenic acid (at 320 nm), kaempferol-3-*O*-rutinoside and kaempferol-3-*O*-robinoside-7-*O*-rhamnoside (at 350 nm), were used to plot calibration curves. Based on this data, quantitative analysis of the polyphenolic constituents in the researched plant was performed. The kaemferol-3-*O*-rutinoside standard was used to determine the monoglycosides of flavonols, the kaempferol-3-*O*-robinoside-7-*O*-rhamnoside standard was used for the remaining flavonols. Chlorogenic acid standard was used to determine hydroxycinnamic acids and protocatechic acid.

### 3.8. Total Reducing Sugar Content

Weighted portions of 1.00 g of researched plant were extracted with 20.0 mL deionized water using a shaker (TTS 2, Yellow Line, IKA-Werke, Staufen, Germany) at room temperature for one hour. Afterwards, supernatant was isolated by centrifugation at 10,000 rpm (MPW-251, Med instrument, Warsaw, Poland) for 15 min and passed through filter paper (BOECO Germany Grade 3 hw).

The volume of 0.5 mL of water extract was mixed with DNS at 1:1 volume ratio and incubated in boiling water for 5 min. The mixture was then cooled under tap water and 4 mL of deionized water was added. The absorbance of the mixture was measured at 540 nm using HP8453 spectrophotometer (Hewlett Packard, Palo Alto, CA, USA/Agilent, Santa Clara, CA, USA, USA). The reducing sugars content in the sample was calculated on the basis of a standard curve prepared for glucose and it was expressed in mg of glucose per g of the plant mass.

### 3.9. Extraction Procedure for Biological Analyses

Weighted portions of 1.00 g dry plant mass were extracted with 10.0 mL of 70% ethanol (*v*/*v*) using a shaker (TTS 2, Yellow Line, IKA-Werke, Staufen, Germany) at room temperature for 24 h. The supernatant was passed through filter paper (BOECO Germany Grade 3 hw). Ethanol was gradually evaporated and the residue was collected and freeze dried. The sample was stored in the refrigerator (4 °C).

### 3.10. Cell Cultures

Human colon adenocarcinoma cell line Caco-2 was obtained from American Type Culture Collection (ATCC, Manassas, VA, USA). Cells were grown in DMEM medium with 10% Fetal Bovine Serum supplemented with 100 U/mL penicillin, 100 µg/mL streptomycin and 25 µg/mL amphotericin B. Cells were cultured at 37 °C in a humidified incubator with 5% carbon dioxide. Tested extract was dissolved in dimethylsulfoxide (DMSO) at a concentration 100 mg/mL and was further diluted with culture medium. The specific concentrations of plant extracts used in the course of various biological studies are presented in the Results section, in respective descriptions of the individual tests carried out. All the experimental measurements were performed using the Synergy 2 BioTek Microplate Reader (BioTek, Winooski, VT, USA). All cell culture reagents were obtained from Life Technologies (Carlsbad, CA, USA), unless stated otherwise.

### 3.11. Cell Viability

Cells in complete medium were seeded in 96-well plates at 10^4^ cells per well and grown for 20 h. Then, cells were incubated for another 24 h in the presence of the studied extracts diluted in culture medium. Cell viability was quantified with PrestoBlue reagent according to the manufacturer’s instructions by measuring the fluorescent signal at F530/590 nm. The obtained fluorescence readings were used to calculate cell viability expressed as a percentage relative to the viability of the control culture (cells treated with equal volume of the vehicle).

### 3.12. Oxidative Stress

To evaluate the protective effect of preparations against oxidative stress, cells were pre-incubated with IC_0_ (concentration at which the viability amounted to 100%) of extracts for 24 h. Then, to induce oxidative stress conditions, 500 μM tert-butylhydroperoxide (*t*-BOOH) (Sigma-Aldrich, St. Louis, MO, USA) was added for 2 h and viability of cells was measured.

To determine the effect of extracts on the intracellular generation of ROS, cells were incubated with extracts for 24 h and afterwards—loaded with DCFH-DA (Sigma-Aldrich, St. Louis, MO, USA) dye at a final concentration of 10 μM for 30 min. Fluorescent signal was analyzed at F485/530 nm.

### 3.13. Mitochondrial Membrane Potential (MMP)

The MMP was assayed with JC-1 probe (Sigma-Aldrich, St. Louis, MO, USA). After 24 h treatment of cells with the studied extracts, 1 μg/mL of JC-1 was added for 20 min. Afterwards, the cells were washed with serum-free medium and fluorescent signal was measured at F485/530 nm. As a known mitochondrial uncoupler and internal positive control, carbonyl cyanide 3-chlorophenylhydrazone (CCCP) was used at 50 μM.

### 3.14. ATP Production

Intracellular ATP level was determined with CellTiter-Glo^®^ Luminescent Cell Viability Assay (Promega Corp., Madison, WI, USA). Briefly, upon cells incubation with the extracts for 24 h, the single reagent was added directly to the cells. Upon cells lysis, luminescent signal proportional to the amount of produced ATP was generated and measured.

### 3.15. Phosphatidylserine Externalization

To quantify the level of phosphatidylserine externalized on the outer leaflet of cell membrane of apoptotic cells, Annexin-V-FITC assay kit (Sigma-Aldrich, St. Louis, MO, USA) was used. After 24 h treatment with the plant extracts, cells were washed twice with PBS and incubated with annexin-V-FITC (final concentration 0.25 μg/mL) for 10 min. Annexin-V binding was measured by the change in fluorescence (excitation/emission = 485/530 nm).

### 3.16. Membrane Permeabilization

Membrane permeabilization of cells caused by the studied plant extracts was measured with propidium iodide (PI) (Sigma-Aldrich, St. Louis, MO, USA). After 24 h treatment, PI was added at a final concentration of 1 µg/mL. Intercalation was monitored by registering the change of fluorescence (excitation/emission = 535/620 nm).

### 3.17. Detection of Caspases 3/7 Activity

The late stage of apoptosis was measured with Caspase 3/7 Assay Kit (Promega Corp., Madison, WI, USA) according to the manufacturer’s instructions. After 24 h treatment with plant extracts, cells were lysed and assay buffer with Ac-DEVD-AMC substrate was added at a final concentration 10 µM. As positive control, 1 µM staurosporine was used. After 30 min incubation, fluorescence was measured (excitation/emission = 360/460 nm).

### 3.18. Statistical Analysis

All data obtained with the use of cell cultures are presented as mean ± SD calculated from at least three independent experiments. Cells in the control sample were exposed only to the vehicle. All obtained results were subjected to statistical analysis using one-way ANOVA analysis, followed by the Dunnett’s test performed using GraphPad Prism 6.0 software (GraphPad Software Inc., La Jolla, CA, USA) at the significance level of * *p* ≤ 0.05, ** *p* ≤ 0.01, *** *p* ≤ 0.001.

## Figures and Tables

**Figure 1 molecules-27-04609-f001:**
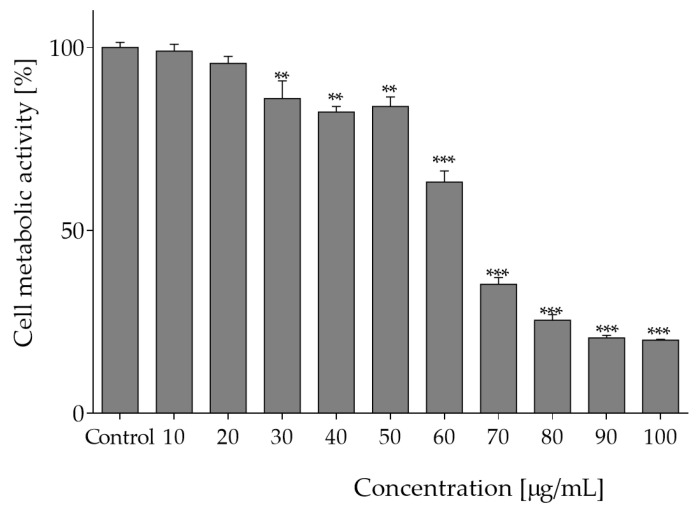
The influence of *O. pseudoglandulosa* extract on Caco-2 cells metabolic activity determined with PrestoBlue assay upon 24 h of incubation. Control cells were only exposed to the vehicle. Values were shown with mean standard deviations, *n* ≥ 9. Statistical significance was calculated against control cell culture (untreated) with ** *p* ≤ 0.005, *** *p* ≤ 0.001.

**Figure 2 molecules-27-04609-f002:**
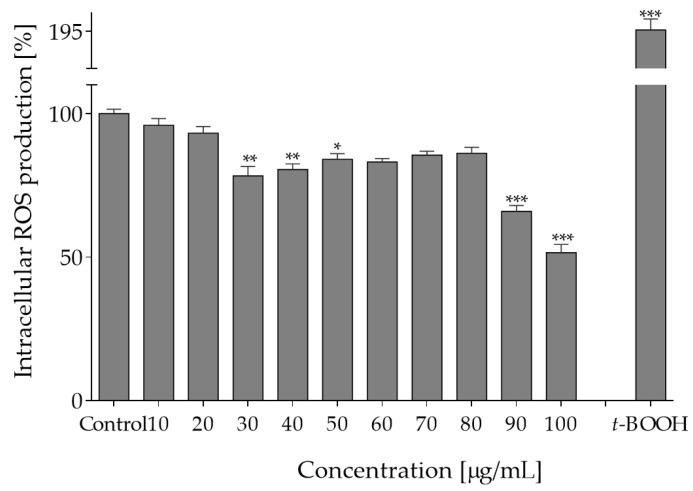
The effect of *O. pseudoglandulosa* extract on forced intracellular ROS generation in Caco-2 cells determined with DCFH-DA assay upon 24 h of incubation. Control cells were only exposed to the vehicle. As a positive control, ROS were determined in cells subjected to 500 µM *t*-BOOH. Values were shown with mean standard deviations, *n* ≥ 9. Statistical significance was calculated against control cell culture (untreated) with * *p* ≤ 0.05, ** *p* ≤ 0.01, *** *p* ≤ 0.001.

**Figure 3 molecules-27-04609-f003:**
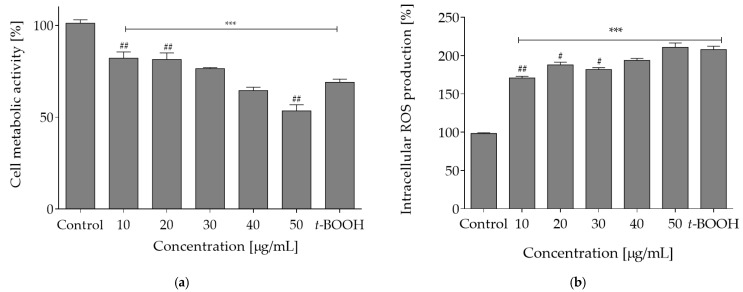
Cytoprotective properties of *O. pseudoglandulosa* determined in Caco-2 cell culture preincubated for 24 h with different plant extract concentrations followed by chemical induction of oxidative stress with 500 μM *t-*BOOH for 2h. Control cells (first bar) were preincubated with the vehicle and not subjected to induced oxidative stress. The last bar represents control cell culture (preincubated with vehicle) subjected to induced oxidative stress. Mean values shown with standard deviations, *n* ≥ 12. Statistical significance was calculated against control cell culture with *** *p* ≤ 0.001 or against *t*-BOOH-treated control cell culture with ^#^ *p* ≤ 0.05, ^##^ *p* ≤ 0.01. (**a**) Intracellular ROS level determined with DCFH-DA assay. (**b**) Viability reflected by changes in cellular metabolic activity determined with PrestoBlue assay.

**Figure 4 molecules-27-04609-f004:**
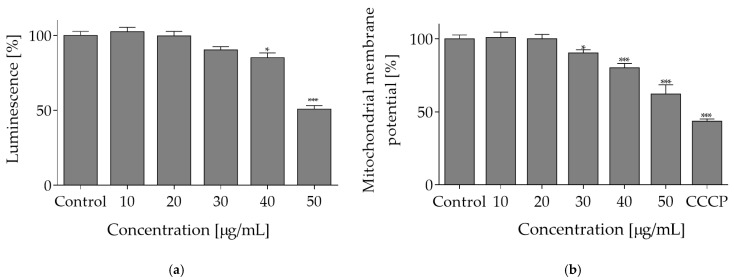
The mechanism of *O. pseudoglandulosa* extract cytotoxicity. (**a**) The influence of *O. pseudoglandulosa* extract on the Caco-2 cells ATP level determined with ATP luminescent assay upon 24 h exposure. (**b**) Mitochondrial membrane potential (MMP) determined with JC-1 probe. Carbonyl cyanide m-chlorophenyl hydrazine (CCCP) at 50 μM was used for cell treatment in a positive depolarization control. Control cells were only exposed to the vehicle. Mean values shown with standard deviations, *n* ≥ 9. Statistical significance was calculated against control cells (untreated) with * *p* ≤ 0.01, *** *p* ≤ 0.001.

**Figure 5 molecules-27-04609-f005:**
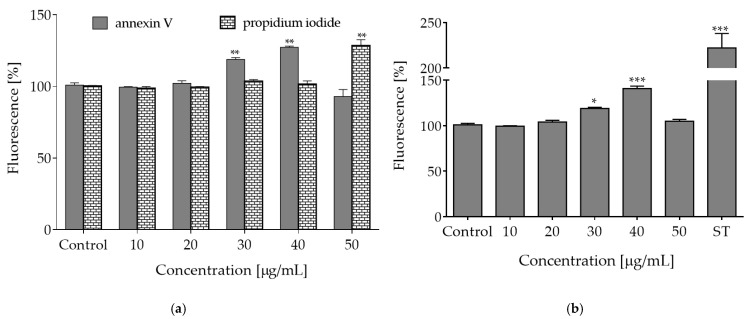
The type of cellular death induced by *O. pseudoglandulosa* extracts. (**a**) The influence of *O. pseudoglandulosa* extract on the Caco-2 cells with regard to phosphatidylserine (PS) externalization on the outer membrane leaflet of apoptotic cells as well as membrane permeabilization of necrotic cells. (**b**) The influence of *O. pseudoglandulosa* extract on caspases-3/7 activation; ST—staurosporine used as a positive control. Control cells were only exposed to the vehicle. Mean values shown with standard deviations, *n* ≥ 9. Statistical significance was calculated against control cells (untreated) with * *p* ≤ 0.01, ** *p* ≤ 0.005, *** *p* ≤ 0.001.

**Table 1 molecules-27-04609-t001:** Identification of polyphenolic compounds in *O. pseudoglandulosa*.

No.	Identification	R_time_	UV Max	[M − H]^−^ *m*/*z*	Fragmentation Ions
1	Protocatechuic acid	13.56	317	315.09	153.02, 152.01
2	5-Caffeoylquinic acid (stn)	18.82	328	353.09	191.06
3	4-Caffeoylquinic acid	19.22	296/320	353.09	191.06,179.03, 173.03
4	K-3-(Rhm)Rut-7-Rhm	21.51	267/347	885.27	739.21, 285.04
5	Q-3-*O*-Rut-7-*O*-Rhm	23.56	259/356	755.21	609.15, 301.03
6	K-3-Rut-7-Rut	24.02	267/347	901.27	593.15, 285.04
7	K-3-*O*-Rob-7-*O*-Rhm (stn)	25.27	267/349	739.21	593.15, 285.04
8	K-3-*O*-Rut-7-*O-*Rhm	25.98	266/347	739.21	593.15, 285.04
9	K-3-(p-Coum, Rhm)Rut-7-Rhm	28.76	270/318	1031.31	885.25, 739.20, 539.15, 285.04
10	K-3-*O*-Rut (stn)	29.19	267/342	593.15	285.04
11	I-3-*O*-Rut	30.57	267/355	623.16	315.05
12	K-3-*O*-Rhm	40.92	267/364	431.10	285.04

Abbreviations: K—Kaempferol, Q—Quercetin, I—Isorhamnetin, *p*-Coum—*p*-coumaroyl residue, Rut—rutinoside, Rhm—rhamnoside, Rob—robinoside, stn—identification confirmed with the standard.

**Table 2 molecules-27-04609-t002:** Quantitative analysis of polyphenolic compounds in *Oxytropis pseudogladulosa*.

No.	Compound	Total Content (mg/kg Dry Mass)
1	Protocatechuic acid	26.7
2	5-Caffeoylquinic acid (std)	34.4
3	4-Caffeoylquinic acid	69.5
4	K-3-(Rhm)Rut-7-Rhm	1592.8
5	Q-3-*O*-Rut-7-*O*-Rhm	525.5
6	K-3-Rut-7-Rut	387.5
7	K-3-*O*-Rob-7-*O*-Rhm (std)	12355
8	K-3-*O*-Rut-7-*O*-Rhm	498.9
9	K-3-(*p*-Coum, Rhm)Rut-7-Rhm	861.4
10	K-3-*O*-Rut (std)	620.2
11	I-3-*O*-Rut	307.7
12	K-3-*O*-Rhm	318.4
	Total polyphenolic compounds	17,598

## Data Availability

Not applicable.

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
