# Peer review of "Chemical Components of Oxytropis pseudoglandulosa Induce Apoptotic-Type Cell Death of Caco-2 Cells"

_molecules, 2022, doi:10.3390/molecules27144609_

Round 1
Reviewer 1 Report
In this study, the total content 14 of polyphenols and flavonoids as well as antioxidant activity were verified in plant extract then biological properties of O. pseudo- 21 glandules extracts were determined in vitro using human epithelial adenocarcinoma Caco-2 cell 22 line. The author wrote the manuscript very clearly and the theory is interesting. Only the introduction part I feel less discussion. Authors can add more discussion related previous study to highlight the importance of the work. The manuscript have some novel finding hence I recommend this article for publication after minor revision.
Author Response
Thank you very much for your review. The comments have been incorporated into the text.
Reviewer 2 Report
Comments to the manuscript molecules-1806215 "Chemical components of Oxytropis pseudoglandulosa induce apoptotic-type cell death of Caco-2 cells".
Authors propose the report of a research aimed to increase the knowledge of the chemical composition of the Oxytropis pseudoglandulosa plant. Moreover, some biological properties of the plant extracts were tested to obtain control of the Caco-2 cells, as well as to study their antioxidant potential. The research was correctly designed and carried out. Results are clearly positive and represent a good contribute to the field of study. The manuscript contains a good state of the art, a complete description of the materials and methods used and a clear discussion of the results. In my opinion, the manuscript is potentially suitable for publication after some minor editing changes. However, my criticism is mainly focused on the bad quality of the Figures representing the observed data. The characters used for the axes titles and labels are too small and not clearly visible. Moreover, the grey colour used for the hystograms is too pale and do not permits a easy evidence of the recorded values. Please improve all the Figures.
Author Response
Thank you very much for your review. Comments have been included in the text - the drawings have been corrected.